**Data Availability Statement:** We have translated the raw transcripts of the interviews and uploaded

# Implementation of Pelvic Floor Rehabilitation after rectal cancer surgery: A qualitative study guided by the Consolidated Framework for Implementation Research (CFIR)

**N. M. Bosch**[1‡], **A. J. Kalkdijk-Dijkstra**[2‡], **P. M. A. Broens**[2], **H. L. van Westreenen**[3], **J. P. E. N. Pierie**[4,5], **B. R. Klarenbeek**[1]*, **J. A. G. van der Heijden**[1,6], on behalf of the FORCE trial group[¶]

1 Department of Surgery, Radboud University Medical Centre, Nijmegen, the Netherlands, 2 Department of Surgery, University Medical Centre Groningen, Groningen, the Netherlands, 3 Department of Surgery, Isala Clinics, Zwolle, the Netherlands, 4 Department of Surgery, Medical Centre Leeuwarden, Leeuwarden, the Netherlands, 5 Department of PGSoM, Medical Centre Leeuwarden, Leeuwarden, the Netherlands, 6 Department of Surgery, Catharina Hospital Eindhoven, Eindhoven, the Netherlands

☯ These authors contributed equally to this work.
‡ NMB and AJKD are joint first authors on this work.
¶ Force trial group, a list is provided in the Acknowledgments.
* Bastiaan.klarenbeek@radboudumc.nl

## Abstract

### Background

Pelvic Floor Rehabilitation (PFR) is effective in a selection of patients with low anterior resection syndrome (LARS) after rectal cancer surgery. This study aimed to identify barriers and enablers to prepare for successful implementation into clinical practice.

### Methods

A qualitative study was performed, guided by the Consolidated Framework for Implementation Research (CFIR). Individual interviews (n = 27) and two focus groups were conducted to synthesize the perspectives of rectal cancer patients, pelvic floor (PF) physiotherapists, and medical experts.

### Results

Barriers were found to be the absence of guidelines about LARS treatment, underdeveloped network care, suboptimal patient information, and expectation management upfront to PFR. Financial status is frequently a barrier because insurance companies do not always reimburse PFR. Enablers were the current level of evidence for PFR, the positive relationship between patients and PF physiotherapists, and the level of self-motivation by patients.

them in DANS easy, an open access data repository. The data is currently being processed and will be available via the DOI: https://doi.org/10.17026/dans-xmk-twgb.

**Funding:** There was no funding for this study, although the original (FORCE) trial itself was funded by the Netherlands Organization for Health Research and Development (ZonMw, file number 80-84300-98-72021). The funder has had no role in the conceptualization, design, data collection, analysis, decision to publish, or preparation of the manuscript.

**Competing interests:** The authors have declared that no competing interests exist.

**Abbreviations:** CFIR, Consolidated Framework for Implementation Research; LARS, Low Anterior Resection Syndrome; PF, Pelvic floor; PFR, Pelvic Floor Rehabilitation; QoL, Quality of Life.

## Conclusion

The factors identified in our study play a crucial role in ensuring a successful implementation of PFR after rectal cancer surgery.

## Introduction

Rectal cancer is a disease in which malignant cells form in the tissues of the rectum. It has a lifetime probability of around 5% and the majority of disease occurs in people older than 50 with a slight predominance in males. Low anterior resection (LAR) is frequently used form of sphincter-preserving surgery and has good oncological outcomes. However, up to 90% of patients after LAR suffer from anorectal dysfunction, which is collectively called the low anterior resection syndrome (LARS) [1–5]. The symptoms associated with this syndrome, which include fecal incontinence, fragmentation, and clustering, significantly impact patient's physical, mental and social functioning [1, 6–9].

In the Cochrane review by Wu et al., it is stated that pelvic floor rehabilitation (PFR) can alleviate LARS complaints and that it is worthy of popularization and application [10]. Recently, the FORCE trial, which is a multicentre randomized clinical trial that investigated whether a structured and predefined pelvic floor rehabilitation (PFR) program is effective in improving postoperative incontinence levels and (fecal incontinence-related) quality of life, found that roughly 85% of patients could benefit from this intervention [11]. This includes patients with urgency or at least moderate incontinence, excluding those with near-complete incontinence.

In the Netherlands, PFR is already covered by basic health insurance for patients with urinary incontinence. However, no literature regarding implementation strategies of PFR for urinary incontinence is available. Additionally, for other indications of PFR, such as dyspareunia, no implementation studies have been published. Gynecology, especially in post-gynecogolocial cancer care, is a field of medicine in which PFR is already being used, and implementation studies have been performed.

Cyr et al. [12] explored the patients' perspective after participation in a multimodal PFR program and found the intervention to be accepted and recommended by all patients, even though therapy was sometimes reported as demanding.

When aiming to implement new treatments, many practical problems may be encountered, such as lack of an adequate training program, resistance from stakeholders, limited resources, or organizational issues. Given that no literature is available that evaluates the implementation of PFR after LAR for rectal cancer, there is no awareness of the potential barriers or enablers for a successful implementation of PFR in daily practice. The aim of this study is therefore to analyze the presence of enablers and barriers influencing the implementation of PFR.

## Methods

### Study design and study setting

This study represents a qualitative follow-up study of the FORCE trial, utilizing the Consolidated Framework for Implementation Research (CFIR) [13] for data collection and analysis. The FORCE trial was registered in the Netherlands Trial Registration (NTR5469) in 09–2015, and was approved by the Medical Ethics Committee Arnhem/Nijmegen in 05–2027 (NL59799.091.16). CFIR is a widely used meta-theoretical framework that assesses the determinants of implementation, encompassing five domains: (1) intervention characteristics, (2)

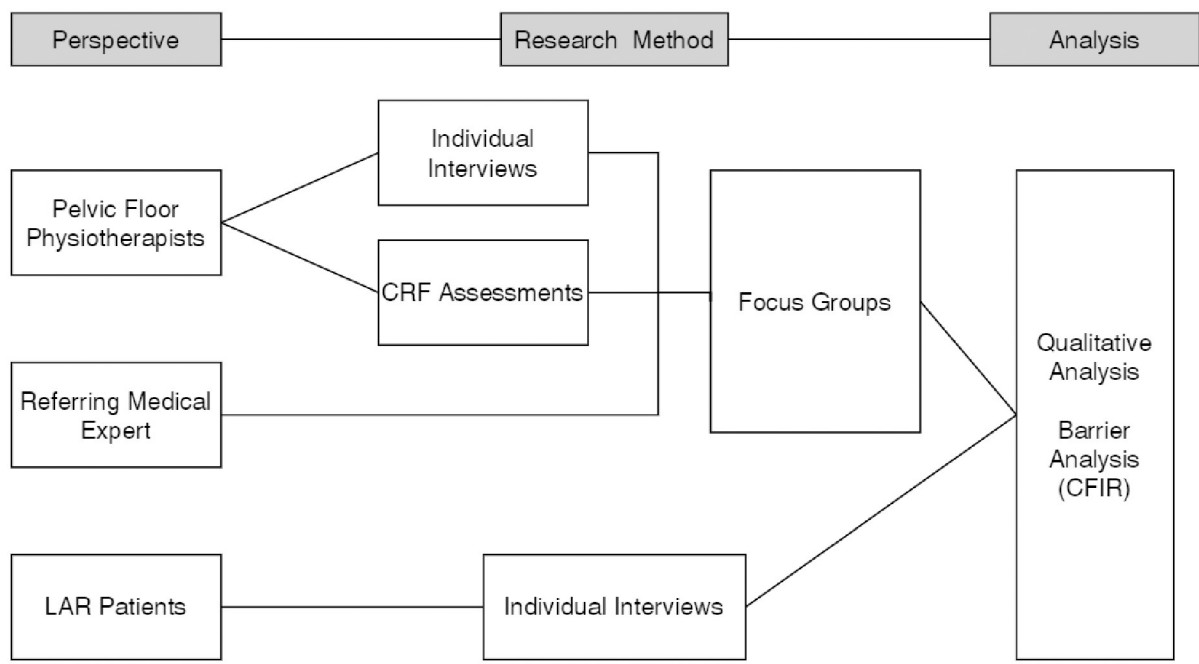

**Fig 1. Flowchart of methods.** Abbreviations: LAR: low anterior resection. CRF: care report file.

inner setting, (3) outer setting, (4) characteristics of individual involved in the implementation, and (5) implementation process. Within these domains, there are 39 CFIR constructs and sub-constructs theoretically associated with the effective implementation of an innovation. The updated CFIR based on used feedback, as published in 2022, was used.

A mixed-method design was employed to evaluate all stakeholders. Individual interviews were conducted with PF physiotherapists and patients who participated in the FORCE trial, while focus groups were organized for PF physiotherapists and medical experts (surgeons, colon care nurses, specialized incontinence nurses, etc.). See Fig 1. The outcomes of the FORCE trial were not yet published or shared with the participants to prevent bias. The individual interviews aimed to gather perspectives on the intervention from its inception until its conclusion. The focus groups aimed to identify implementation barriers and facilitate consensus among participants regarding the findings of the individual interviews by encouraging the participants to discuss the topics amongst themselves [14, 15]. On average, individual interviews lasted 45 minutes, while focus groups lasted 90 minutes. Based on the aforementioned findings, recommendations for clinical practice were formulated, drawing from the considerations of Eccles et al. [16, 17], provides a well-considered approach for the evaluation and reporting of recommendations.

Three investigators (NB, JK, JH) conducted the semi-structured interviews. The questions were designed to align with the CFIR domain and address the current status of patient referral to PFR, collaboration between medical experts and physiotherapists, and expectations regarding the duration, effects, and final reports at the end of the treatment. In addition, the individual interviews with patients explored their personal experiences regarding the intervention, the value of the aftercare and follow-up, contact and cooperation with healthcare workers, and financial consequences.

Due to covid-19 restrictions, all sessions were conducted online. Specific preparations were taken to ensure that the virtual focus groups yielded similar outcomes as in-person sessions

(adherence to online meeting rules, video connections, active participation of the attendees) [18]. The focus group sessions were facilitated by a moderator (JH) with previous experience in qualitative research.

## Study population

Three different perspectives were explored. Firstly, PF physiotherapists. At least one physiotherapist affiliated with each trial-participating hospital (since it was a multi-center study) was invited to participate. Secondly, referring medical experts who regularly follow up with patients after sphincter-preserving rectal cancer surgery. All medical experts who were part of the FORCE trial collaborating group were eligible to participate in the focus groups. The aim was to have a minimum of two medical experts per focus group. Thirdly, patients who underwent PFR after rectal cancer surgery. Three subgroups, each consisting of an equal number of patients, were invited. These subgroups included patients who showed above-average improvement in incontinence scores after PFR, those who showed below-average improvement (or even deterioration), and patients who were assigned to the control group but chose to undergo PFR after completing the trial. The coordinating researcher randomly approached four to eight patients from each group and asked them to participate in one-on-one interviews conducted by an independent researcher.

## Data analysis

The audio recordings of the interviews and focus groups were transcribed and analyzed by two investigators: the focus group moderator (JH) and a second investigator (NB). In case of disagreement, a third researcher was counseled (JK). Coding was performed based on the initial assessments and impressions obtained from listening to and reading the transcripts. Data saturation was achieved when consensus was reached, and no new information emerged in subsequent focus groups.

## Results

In total, thirteen PF physiotherapists, six surgeons, and one continence care nurse participated. Twelve physiotherapists participated in the individual interviews. Five physiotherapists, six surgeons, and one incontinence care nurse participated in the focus groups. Characteristics of the participating physiotherapists and medical experts are provided in Table 1. Most of them were female (90%). The median age of the participants was 47 years (IQR 39–49). The majority

**Table 1. Characteristics of the Dutch PF physiotherapists and medical referrers.**

| | | PF physiotherapists n = 13 | Medical experts n = 7 |
|---|---|---|---|
| **Participation** | Individual interviews | 12 | 0 |
| | Focus group | 6 | 7 |
| **Care** | Primary care | 11 | 0 |
| | Secondary care | 2 | 7 |
| **Experience (LAR patients/year)** | <5 | 3 | NA |
| | 5–10 | 5 | NA |
| | >10 | 4 | NA |
| **Sex** | Female | 13 (100%) | 5 (71.4%) |
| | Male | 0 | 2 (28.6%) |

Abbreviations: NA: not applicable. LAR: low anterior resection. PF: pelvic floor. N = number.

of the participating physiotherapists worked in primary care (84.6%), one in secondary care (7.7%), and one in both (7.7%). All participating PF physiotherapists had experience with the treatment of patients after rectal cancer; two participants (15.4%) over ten patients per year, and twelve participants (84.6%) 5–10 per year. In total, fifteen of the 55 PFR program participants in the FORCE trial participated in individual interviews. Seventeen were asked for participation, of which one developed metastasis and refused participation and one did not want to continue participation in another scientific research project.

### #1 Innovation

**Enablers.** *Construct*: *Evidence*. PFR has a positive effect on the QoL for the majority of patients. There is a present need for the implementation of PFR. According to the patients, they didn't expect such a positive effect on their QoL due to PFR. They had more insight into their symptoms and experienced improved awareness of their pelvic floor. This was perceived by both the physiotherapists and the patients who received therapy. Patients emphasized public interest in the nationwide implementation of PFR after rectal surgery.

*Construct*: *Advantage*.

1. Better informed patients due to the logistics of the trial protocol.
   According to physiotherapists, patients who were referred for PFR outside the context of the FORCE trial were usually unprepared, while patients who followed the trial logistics were up-front and adequately informed with better expectations of the intervention, this was by means of patient information by digital and written content (S1 File). 2) The currently present guideline 'fecal incontinence' does not cover symptomatology and treatment of LARS. Physiotherapists did not feel supported by the current guideline. There is a need for evidence based-guidelines for the treatment of LARS symptoms.

*Construct*: *Design*. Presence of a standardized PFR program derived from the FORCE trial. According to the physiotherapists and medical experts, the trial protocol was perceived as feasible and suggested as the backbone for the intervention.

**Barrier.** *Construct*: *Design*. Reported concerns regarding the frequency, sequence, and content of treatment sessions. Some changes in treatment protocol were suggested, such as to start with focusing on awareness training and educational training prior to (invasive) baseline measurements which was in accordance with the preference of the majority of the patients. Physiotherapists thought a treatment frequency of once a week is reasonable; however, it is not suited for twelve successive weeks. Weekly sessions were suggested for the first 6 weeks, after that a longer interval was recommended in order to allow prolonged guidance and therefore possibly better results. The participating patients also favored this.

### #2 Outer setting

**Barrier.** *Construct*: *Financing, politics & law*. *Lack of reimbursement of PFR by insurance*. By participating in the trial, PFR was financially compensated by research funding. Otherwise, PFR for LARS or fecal incontinence is (in the Netherlands) not always reimbursed by insurance companies. Physiotherapists mentioned patients being withheld from PFR or unable to finish therapy due to their lack of financial resources. All stakeholders agreed that reimbursement is of the utmost importance to allow access to care and provide uniform treatment possibilities.

### #3 Inner setting

**Enablers.** *Construct*: *Communication*.

1. Positive acknowledgment of communication between patients and PF physiotherapists. Their communication was experienced very pleasantly, improving patients' motivation and how they perceive the intervention. Most patients appreciated physiotherapists explaining the progress and managing the expectations during the program. Additional enablers were time availability for questions and attention to the psychosocial aspect.

2. Consensus regarding referral information. A consensus between the physiotherapists and the medical experts was reached regarding the contents of a referral. A standard form with a minimal set of items in a referral letter has been suggested and is added as S2 File.

*Construct*: *Physical and work infrastructure*. Both medical experts and PF physiotherapists were willing to accept PFR in their treatment protocol for LARS complaints. Physiotherapists were willing to implement the trial protocol as the backbone for a multimodal treatment program. In addition, they predicted that to have the capacity in their practices to adequately treat the new influx of patients who are eligible for PFR.

*Construct*: *Culture*. Patients received aid in psychosocial difficulties, such as the opportunity to talk about embarrassing issues (incontinence, urgency, frequent bowel movement issues, etcetera). This helps in patients' motivation and willingness to continue participation.

**Barriers.** *Construct*: *Communication*. Lack of communication networks between hospitals and regional PF physiotherapists. All participants emphasized the importance of regional networks between physiotherapists and surgeons. When such a network is present, communication is often fast and accessible. Participants reported that because of the trial logistics, their regional networks improved already, showing the feasibility of achieving such network care.

*Construct: Culture, recipient-centeredness, and deliverer- and learning-centeredness.*

1. The need for better expectation management from the referring party regarding PFR. Some patients reported that they needed better expectation management with regard to the intervention.

2. Maintaining a level of expertise for PF physiotherapists. Both physiotherapists and medical experts were in unison that only physiotherapists with expertise in LARS and post-rectal cancer surgery anatomy should treat patients. To overcome the learning curve, additional education or training is desired. Predominantly information about the medication, nutrition, change in anatomy, impact of radiotherapy, and the rectal surgery itself.

## #4 Individual

**Enablers.** *Construct*: *Mid-level leaders*. Physiotherapists were enthusiastic and had a great willingness to be skilled and to progress the implementation of the PFR into practice.

*Construct*: *Innovation recipients*. Level of self-motivation by patients and desire to improve their health.

**Barrier.** *Construct*: *Innovation recipients*. Misinformation, lack of coordination by the treatment team, and feelings of shame (when healthcare workers are not able to alleviate these feelings properly) can be barriers that withhold patients from participation.

## #5 Implementation process

**Enablers.** *Construct*: *Planning*. The medical experts and physiotherapists who are actively engaged possess a comprehensive understanding of the delineation of duties and exhibit a willingness to support and integrate it within their professional activities.

Results from the interviews and focus groups are summarized in Table 2. They are categorized according to the CFIR, and their construct is supported by illustrative quotes derived from the interviews of focus group sessions.

## Discussion

Using a qualitative approach including all relevant stakeholders, we identified a comprehensive set of barriers and enablers that could affect PFR implementation in daily care for patients suffering from LARS complaints after sphincter-preserving rectal cancer surgery. Our findings suggest that successful PFR implementation requires addressing barriers at multiple levels, including the absence of guidelines about LARS treatment, suboptimal patient information and expectation management upfront to PFR, the fact that network care is an important condition for success and at last that financial reimbursement is an enabler when present, though most of the time a barrier since insurance companies do not cover PFR for the indication being discussed. These findings play a crucial role in ensuring the successful implementation of the rehabilitation program.

A perceived barrier was the absence of guidelines for PFR focused on LARS complaints, including the variety of modalities used. Especially since stakeholders reported that existing guidelines focusing on fecal incontinence are not sufficient. Based on the responses in this study, this barrier can be overcome by implementing a treatment protocol, such as being used in the FORCE trial. By doing so, structured use of all four PFR modalities (i.e. pelvic floor muscle training, biofeedback, electrostimulation and rectal balloon treatment) is guaranteed. Patients, however, reported reservations about the use of invasive treatment modalities in the first treatment sessions. An issue that can be overcome by creating awareness of pelvic floor muscles first and mainly by managing their expectations upfront.

Patent-mediated interventions, such as the availability of digital or written patient information/brochures prior to referral and start of therapy [19], are suggested to be used to educate and better engage patients to participate and be well informed prior to the start [20]. It was reported to be important that medical experts also inform patients about the safety of using rectal probes since it is an essential part of PFR. Because both patients and physiotherapists reported the information flyer that was used in the FORCE trial to be adequate, we have added a translated version in the appendices (S1 File).

Previous research conducted in a gynecological setting identified serious barriers to implementing PFR, including misinformation, lack of coordination with the treatment team, and feelings of shame [21].

Building upon these findings, stakeholders in this study reported a lack of adequate information about patients at the time of referral, as well as insufficient communication during therapy. Patients were sent away for therapy, and medical experts did not know whether therapy was successful or stopped prematurely (and if so, for which reasons). Based on these observations, we concluded that establishing a network between local hospitals and PF physiotherapy centers is essential for successful implementation. Such network care is known to reduce costs, improve clinical care quality and provide access to a broad range of healthcare services for all stakeholders involved [22–24].

Based on the suggestions of study participants, network care in the PFR setting should consist of face-to-face/online interactions supplemented by fast communication tools such as email or secure messaging apps. Indications for referral should be discussed, as well as the preferred manner of feedback. We created a backbone for a standardized referral form which is composed of relevant information which, most of the time, cannot be provided by the patient themselves, including information about the type of surgery, the height of the anastomose,

**Table 2.  CFIR domains and constructs.**

| CFIR domain | CFIR construct | Illustrative Quotes |
|---|---|---|
| Implementation outcome | | |
| **I. Innovation** | Evidence | *I didn't expect such a positive effect on my life due to PFR.*<br>• *Patient* |
| | Advantage | *Better-informed patients experience an increase in acceptance of their complaints. They positively influence the way they experience PFR.*<br>• *Physiotherapist* |
| | Design, Adaptability | *The trial protocol should be used for PFR, although more time investment aimed on awareness and the impact on daily functioning is needed in the first sessions, in addition these sessions could be used for more information regarding the contents of PFR.*<br>• *Physiotherapist* |
| | Design, Adaptability | *Twelve weekly training sessions is not practical outside the study construct. For the first 6 sessions a higher frequency (1x/week) is necessary in order to teach awareness of pelvic floor muscles and getting used to the tools (i.e. rectal probe). After that, muscles need to be strengthened over time which can be better achieved with a lower frequency of sessions. It also prolongs the total time of guidance.*<br>• *Physiotherapist* |
| | Advantage | *The Guideline fecal incontinence does not cover the symptoms and treatment of LARS and we experience not enough support by this guideline. There should be an evidence-based guideline for diagnosis and treatment of symptoms of LARS.*<br>• *Physiotherapist* |
| **II. Outer setting** | Financing, Politics & Laws | *Similar to PFR for urinary incontinence, it should be reimbursed for patients with post-operative rectal dysfunction as well.*<br>• *Medical expert* |
| | Financing, Politics & Laws | *It is not okay that urinary incontinence has 9 treatment sessions included in basic health insurance and fecal incontinence does not, while the impact [of fecal incontinence] is greater. The costs are sometimes not affordable for patients, and then I try to think about other ways to treat them which is way less effective. This should not be necessary, PFR should be compensated.*<br>• *Physiotherapist* |
| **III. Inner setting** | Relational connections | *Before the intervention, after referral I completely lost sight of the patient. I don't know to which physiotherapist they go and where to send my referral information to. It is like there is a black box where the patients disappear into.*<br>• *Medical expert* |
| | Relational connections | *The FORCE trial already improved my network between the hospital and the physiotherapists in my region, so it is feasible to create these networks fast.*<br>• *Medical expert* |
| | Relational connections | *Low-threshold contact is necessary for better healthcare, thus everyone has to put energy in creating that network.*<br>• *Medical expert* |
| | Relational connections | *We have short lines of communication with a few certified pelvic physiotherapists in the region with whom we regularly consult about new treatment options for example.*<br>• *Medical expert* |
| | Incentive systems, Available resources | *We have the capacity and willingness to provide the postoperative rectum cancer patients with PFR in our facility. And otherwise, the capacity should be increased to allow these new patients to be treated.-*<br>• *Physiotherapist* |
| | Communication | *The referral letter from surgeon to physiotherapist must contain a minimal set of items. Most of the time it is too brief or incomplete.*<br>• *Physiotherapist* |
| | Culture, Deliverer- and Learning-Centeredness | *Not every physiotherapist should treat patients with functional complaints after rectum cancer. There must be a certain level of expertise and level of knowledge. It should be practice by physiotherapist specialized in pelvic floor rehabilitation after surgery.*<br>• *Medical expert*<br>• *Physiotherapist* |
| | Culture, Recipient-Centeredness | *It is helpful to talk about these complaints [of fecal incontinence, urgency, frequent bowel movements] which are often difficult to discuss. My physiotherapist made it a more comfortable subject.*<br>• *Patient* |

*(Continued)*

**Table 2.** (Continued)

| CFIR domain | CFIR construct | Illustrative Quotes |
|---|---|---|
| IV. Characteristics of individuals | Mid-level leaders | *Physiotherapists are enthusiastic and willing to do everything to be skilled and up to date with existing literature for this treatment. We will support implementation of the treatment.*<br>• *Physiotherapist* |
| V. IMPLEMENTATION PROCESS | Teaming | *Medical experts should actively direct patients to PFR and coordinate together with physiotherapists.*<br>• *Patient* |

previous complications, and medical restriction for the use of an anal probe or rectal balloon (S2 File).

At the end of therapy, feedback from the attending physician about the effect of treatment on a functional level, psychosocial development, and expectations for future improvements does not always reach the referring parties. Previous research shows that such feedback is an enabler when present or a barrier when absent [21].

At last, the treatment costs of PFR were mostly perceived as a barrier since PFR is (in the Netherlands) not covered by basic health insurance for the indication of LARS. Given that many patients lack the financial recourses to reimburse treatment themselves, health insurance will play a critical role in preventing this barrier. For health insurance companies, concepts such as necessity, effectivity, social perspective, cost-effectiveness and feasibility are important. The present study explored the necessity and the perspectives of the stakeholders involved and confirms the feasibility of implementing a PFR protocol such as being used in the FORCE trial. Considering that as of 2006, nine sessions of PFR for urinary are covered by basic health insurance in the Netherlands [25, 26], we suggest considering coverage PFR for LARS complaints and fecal incontinence too, since it is generally accepted that these issues have a greater impact on QoL and psychological wellbeing than urinary incontinence [27] and enough evidence is available to support effectiveness when selection of patients is made right [10, 11].

Study limitations are similar to many implementation research projects. Because recruitment for interviews and focus groups was voluntary, those agreeing to participate may have had positive experiences or were otherwise motivated to participate, even though we included patients that did not objectively improve in incontinence scores after PFR. As such, the experiences of stakeholders who are unwilling to undergo practice changes may be unheard.

Another important issue is that current evaluation is predominantly focused on the system around the patient, while the patient itself is the most important factor for the success of the therapy. Based on previous research in a different setting where PFR was used (post gynecological oncology care), the level of self-motivation, desire to improve their health, availability of time, support of their partner, and symptoms of improvement over time were all relevant enablers [21]. At last, the relatively small number of participants is considered a limitation, although all participants had recent experiences in PFR and were able to provide useful and up-to-date information.

This paper is unique in its use of implementation theories, including the use of the CFIR framework, to understand clinical practice change in this field of care and also in the intervention that is investigated. The engagement of relevant stakeholders, including patients themselves, leads to reliable outcomes. Therefore, the findings will in the end help to guide PFR implementation within the field of functional complaints after rectal cancer surgery.

## Conclusion

The identified barriers and enablers to the implementation of Pelvic Floor Rehabilitation after rectal cancer treatment have been gathered from the viewpoints of pelvic floor physiotherapists, medical experts, and patients who have undergone post-rectal cancer treatment. These findings play a crucial role in ensuring the successful implementation of the rehabilitation program.

## Supporting information

**S1 File. Patient information letter prior to referral to PFR, as provided by the FORCE trial.**
(DOCX)

**S2 File. A list of items suggested to be included in the referral letter to PFR.**
(DOCX)

## Acknowledgments

The authors acknowledge Prof. dr. P.J. van der Wees (IQ Healthcare, Radboud University Medical Centre, Nijmegen) for his support in the conceptualisation of this study. The authors also acknowledge the surgery departments, research assistants and the participating physicians, nurses, and pelvic floor physiotherapists of the participating hospitals and Pelvic Floor Physiotherapy centers for carrying out this study: FORCE Trial Group. The FORCE trial group consists of all principal investigators, including surgeons, research coordinators, research/oncology/ gastroenterology nurses, colorectal case managers, and participating pelvic floor physiotherapists. For the complete list of participants, we refer to the primary results article (DOI: 10.1097/SLA.0000000000005353). FORCE trial (Netherlands Trial Registration, NTR5469, registered on 3 September 2015).

## Author Contributions

**Conceptualization:** N. M. Bosch, A. J. Kalkdijk-Dijkstra, P. M. A. Broens, H. L. van Westreenen, J. P. E. N. Pierie, B. R. Klarenbeek, J. A. G. van der Heijden.

**Data curation:** N. M. Bosch, A. J. Kalkdijk-Dijkstra, H. L. van Westreenen, J. P. E. N. Pierie, B. R. Klarenbeek, J. A. G. van der Heijden.

**Formal analysis:** N. M. Bosch, A. J. Kalkdijk-Dijkstra, J. A. G. van der Heijden.

**Funding acquisition:** A. J. Kalkdijk-Dijkstra, P. M. A. Broens, H. L. van Westreenen, J. P. E. N. Pierie, B. R. Klarenbeek.

**Investigation:** N. M. Bosch, A. J. Kalkdijk-Dijkstra, P. M. A. Broens, H. L. van Westreenen, J. P. E. N. Pierie, B. R. Klarenbeek, J. A. G. van der Heijden.

**Methodology:** N. M. Bosch, A. J. Kalkdijk-Dijkstra, P. M. A. Broens, H. L. van Westreenen, J. P. E. N. Pierie, B. R. Klarenbeek, J. A. G. van der Heijden.

**Project administration:** N. M. Bosch, B. R. Klarenbeek, J. A. G. van der Heijden.

**Resources:** P. M. A. Broens, H. L. van Westreenen, J. P. E. N. Pierie.

**Software:** J. A. G. van der Heijden.

**Supervision:** P. M. A. Broens, H. L. van Westreenen, J. P. E. N. Pierie, B. R. Klarenbeek, J. A. G. van der Heijden.

**Validation:** B. R. Klarenbeek.

**Visualization:** J. A. G. van der Heijden.

**Writing – original draft:** N. M. Bosch, A. J. Kalkdijk-Dijkstra, J. A. G. van der Heijden.

**Writing – review & editing:** P. M. A. Broens, H. L. van Westreenen, J. P. E. N. Pierie, B. R. Klarenbeek, J. A. G. van der Heijden.

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
