## [Decision Letter · Decision Letter 0]

5 Oct 2023

PONE-D-23-24301Implementation of Pelvic Floor Rehabilitation after rectal cancer surgery: A qualitative study guided by the Consolidated Framework for Implementation Research (CFIR)PLOS ONE

Dear Dr. van der Heijden,

Thank you for submitting your manuscript to PLOS ONE. After careful consideration, we feel that it has merit but does not fully meet PLOS ONE’s publication criteria as it currently stands. Therefore, we invite you to submit a revised version of the manuscript that addresses the points raised during the review process.

We look forward to receiving your revised manuscript.

Kind regards,

Shabnam ShahAli, Ph.D.

Academic Editor

PLOS ONE

2. Please ensure you have included the registration number for the clinical trial referenced in the manuscript.

Reviewers' comments:

Reviewer's Responses to Questions

**Comments to the Author**

1. Is the manuscript technically sound, and do the data support the conclusions?

Reviewer #1: Yes

Reviewer #2: Yes

Reviewer #3: Yes

2. Has the statistical analysis been performed appropriately and rigorously? 

Reviewer #1: Yes

Reviewer #2: Yes

Reviewer #3: Yes

3. Have the authors made all data underlying the findings in their manuscript fully available?

Reviewer #1: Yes

Reviewer #2: Yes

Reviewer #3: Yes

4. Is the manuscript presented in an intelligible fashion and written in standard English?

Reviewer #1: Yes

Reviewer #2: Yes

Reviewer #3: Yes

5. Review Comments to the Author

Reviewer #1: Thank you for providing this opportunity for me to review this good study. This research work is very well executed. However this research was conducted between October 2017 and March 2020. In 2022, an updated version of the CFIR has been released with relatively large changes based on feedback received from CFIR users. According to the publication of this article, it does not have the necessary aspect of novelty in this journal.

Reviewer #2: Comments to the Author

Thank you for the opportunity to review this manuscript. The manuscript reports findings of the Implementation of Pelvic Floor Rehabilitation after rectal cancer surgery: A qualitative study guided by the Consolidated Framework for Implementation Research (CFIR). This is a well conducted qualitative study in an important clinical area. The authors describe a robust, well-designed study and have reported comprehensive results. That said, the manuscript in its present form is accepted without any general comments for consideration. Please could the authors include:

- In the introduction, please add the definition of rectal cancer, prevalence, sex, and age that are most involved. Additionally, explain the low anterior resection syndrome and the sequences.

- Arrange for a review of the written English as there are some instances where the quality of the writing could be improved.

- Please add the name of the ethics committee that provided approval for the study and the date and year approval was obtained in the method section.

Reviewer #3: Abstract:

1. In line 38, Conclusion should be written based on the obtained results. Your results have not been written with certainty.

2. In keywords quality of life should be deleted.

3. In line 30 delete mixed from qualitative research.

Methods

4. Explain how do deepen responses post interview guide questions?

5. In method part, how do you ensure maximum variation, confirmability, trustworthiness of your study?

Data analysis

How do you analyze the data and extract codes?

6. PLOS authors have the option to publish the peer review history of their article (what does this mean?). If published, this will include your full peer review and any attached files.

Reviewer #1: **Yes: **holakoo Mohsenifar

Reviewer #2: **Yes: **Mehrnaz Kajbafvala

Reviewer #3: No

---

## [Author Response · Author response to Decision Letter 0]

2 Dec 2023

Response to Editorial Board

Question: 1. Please ensure you have included the registration number for the clinical trial referenced in the manuscript.

Answer: This registration number is added in the manuscript. 

Question: 2. We note that the grant information you provided in the ‘Funding Information’ and ‘Financial Disclosure’ sections do not match. When you resubmit, please ensure that you provide the correct grant numbers for the awards you received for your study in the ‘Funding Information’ section.

Answer: There was no funding for this study, although the original (FORCE) trial itself was funded by the Netherlands Organization for Health Research and Development (ZonMw, file number 80-84300-98-72021). The funder has had no role in the conceptualization, design, data collection, analysis, decision to publish, or preparation of the manuscript. 

Somehow I cannot change this in the digital manuscript submission system, which is why I have changed this in the manuscript. Hope this is sufficient for you at this moment in time. 

Question 3. Thank you for updating your data availability statement. You note that your data are available within the Supporting Information files, but no such files have been included with your submission. At this time we ask that you please upload your minimal data set as a Supporting Information file, or to a public repository such as Figshare or Dryad. Please also ensure that when you upload your file you include separate captions for your supplementary files at the end of your manuscript. As soon as you confirm the location of the data underlying your findings, we will be able to proceed with the review of your submission.

Answer: The data from is now fully available via an online repository. https://doi.org/10.17026/dans-xmk-twgb

Question: 4. Please include your tables as part of your main manuscript and remove the individual files. Please note that supplementary tables (should remain/ be uploaded) as separate ""Supporting Information"" files

Answer: We have adjusted this. 

Question: 5. Please upload a copy of Figure 1 which you refer to in your text. Or if the figure is no longer to be included as part of the submission please remove all reference to it within the text.

Answer: We have adjusted this. 

 

Response to Reviewers

Question/request. 

Answer: We have made several adjustments to adhere to the style requirements. 

Question/request. 

2. Please ensure you have included the registration number for the clinical trial referenced in the manuscript.

Answer: The registration number for the clinical trial is now referenced in the methods section. 

Question/request. 

Answer: we have adjusted this. 

Question/request. 

Answer: The ethics statement is removed from the ‘declarations’ section and now appears only in the methods section of our manuscript. 

Question/request. 

Answer: We did not have to make changes. 

 

Reviewers' comments:

Comments to the Author

Qestion/request: 

Reviewer #1: Thank you for providing this opportunity for me to review this good study. This research work is very well executed. However this research was conducted between October 2017 and March 2020. In 2022, an updated version of the CFIR has been released with relatively large changes based on feedback received from CFIR users. According to the publication of this article, it does not have the necessary aspect of novelty in this journal.

Answer: Dear reviewer 1, you are correct. The research in which the participants were included was conducted between October 2017 and March 2020. However, the final version of this manuscript (which is a follow-up study of the original trial which ended inclusion in March 2020) already used the updated version of the CFIR as described in the article from Damschroder et al, 2022 (implementation science). We hope that this statement is sufficiently for you. Additionally we have added the following sentence in the manuscript methods section: 

“The updated CFIR based on used feedback, as published in 2022, was used.”

Reviewer #2: Comments to the Author

Thank you for the opportunity to review this manuscript. The manuscript reports findings of the Implementation of Pelvic Floor Rehabilitation after rectal cancer surgery: A qualitative study guided by the Consolidated Framework for Implementation Research (CFIR). This is a well conducted qualitative study in an important clinical area. The authors describe a robust, well-designed study and have reported comprehensive results. That said, the manuscript in its present form is accepted without any general comments for consideration. Please could the authors include:

Question/request: - In the introduction, please add the definition of rectal cancer, prevalence, sex, and age that are most involved. Additionally, explain the low anterior resection syndrome and the sequences.

Answer: We have adjusted the introduction section and added the requested items. 

Please see the revised manuscript. For the selection that is revised, see text below: 

“Rectal cancer is a disease in which malignant cells form in the tissues of the rectum. It has a lifetime probability of around 5% and the majority of disease occurs in people older than 50 with a slight predominance in males. Low anterior resection (LAR) is frequently used form of sphincter-preserving surgery and has good oncological outcomes. However, up to 90% of patients after LAR suffer from anorectal dysfunction, which is collectively called the low anterior resection syndrome (LARS). The symptoms associated with this syndrome, which include fecal incontinence, fragmentation, and clustering, have a significant impact on patient’s physical, mental and social functioning.”

Question/request: - Arrange for a review of the written English as there are some instances where the quality of the writing could be improved.

Answer: We performed an additional written English check and improved the text. 

Question/request: Please add the name of the ethics committee that provided approval for the study and the date and year approval was obtained in the method section.

Answer: We have adjusted the method section and added the requested items. 

Please see the revised manuscript. For the selection that is revised, see text below: 

“The FORCE trial was registered in the Netherlands Trial Registration (NTR5469) in 09-2015, and was approved by the Medical Ethics Committee Arnhem/Nijmegen in 05-2027 (NL59799.091.16).”

Reviewer #3: 

Abstract:

Question/request: 1. In line 38, Conclusion should be written based on the obtained results. Your results have not been written with certainty.

Answer: Dear reviewer 3, it is not clear to me/us in what way we should revise our manuscript based on your statement. Can you perhaps rephrase the question? We are more than happy to address the raised questions. 

Question/request: 2. In keywords quality of life should be deleted.

Answer: We have deleted the keyword Quality of life. 

Question/request: 3. In line 30 delete mixed from qualitative research.

Answer: We have deleted the requested word. 

Methods

Question/request: 4. Explain how do deepen responses post interview guide questions?

Answer: Dear reviewer 3, again it is not completely clear to us what you mean by the abovementioned question. After carefully studying the methods section, is think that you are aiming at the section cited below: 

The focus groups aimed to identify implementation barriers and facilitate consensus among participants regarding the findings of the individual interviews by encouraging the participants to discuss the topics amongst themselves

The interviews (one-on-one) were held before the focus groups. If many questions were to arise from the individual interviews, than the focus groups were used to facilitate consensus (based on the questions that were derived from the interviews). 

We hope that this answers your question sufficiently. 

Question/request: 5. In method part, how do you ensure maximum variation, confirmability, trustworthiness of your study?

Answer: Thank you for this question. The maximum variation was ensured by exploring all the possible perspectives (physiotherapists, referring medical experts, and patients who participated in the intervention themselves). To go a step further and ensure maximum variation within the patients that participated (to prevent selection bias; so that only the patients with success from the intervention would present their opinions in this follow-up study); we randomly selected patients from all ranges of outcome effects (those with a good response to therapy, those with no change, those with deterioration of function over time). Therefore, we are convinced that we did everything in our power to ensure maximum variation. 

Regarding confirmability; data were checked and rechecked throughout data collection and analysis. Not two, but three reviewers were used to ensure that results would likely be repeatable by others. 

Regarding trustworthiness; we can only report an the degree of confidence in data, interpretation and methods as described in our manuscript to ensure the quality and therefore trustworthiness of this study. In the most ideal situation, there would be multiple trials that could evaluate their intervention using CFIR and than compare the results in order to enhance trustworthiness. However, since this is the first RCT within this field of research that performed an qualitative analysis with CFIR; this is not possible. 

We hope that this answers your questions sufficiently. 

Data analysis

Question/request: How do you analyze the data and extract codes?

Answer: 

After transcribing the audio recordings, major themes (codes) were identified. These analyses was done by multiple members of the author group. Afterwards, we organized the data by question and theme by highlighting questions, making notes and adding comments throughout the transcript. These quotes are also used in the tables included in the manuscript, to enhance certain statements. At last, disagreements were solved by means of an extra researcher who evaluated the transcripts. 

Since we believe that this is all summarized in our data analysis section, we have not made any changes. We hope that this answers your question sufficiently.

---

## [Decision Letter · Decision Letter 1]

18 Mar 2024

Implementation of Pelvic Floor Rehabilitation after rectal cancer surgery: A qualitative study guided by the Consolidated Framework for Implementation Research (CFIR)

PONE-D-23-24301R1

Dear Dr. van der Heijden,

We’re pleased to inform you that your manuscript has been judged scientifically suitable for publication and will be formally accepted for publication once it meets all outstanding technical requirements.

Kind regards,

Sagar Panthi, MBBS

Academic Editor

PLOS ONE

Additional Editor Comments (optional):

Reviewers' comments:

Reviewer's Responses to Questions

**Comments to the Author**

1. If the authors have adequately addressed your comments raised in a previous round of review and you feel that this manuscript is now acceptable for publication, you may indicate that here to bypass the “Comments to the Author” section, enter your conflict of interest statement in the “Confidential to Editor” section, and submit your "Accept" recommendation.

Reviewer #1: All comments have been addressed

Reviewer #2: All comments have been addressed

Reviewer #4: All comments have been addressed

2. Is the manuscript technically sound, and do the data support the conclusions?

Reviewer #1: Yes

Reviewer #2: Yes

Reviewer #4: Yes

3. Has the statistical analysis been performed appropriately and rigorously? 

Reviewer #1: Yes

Reviewer #2: Yes

Reviewer #4: N/A

4. Have the authors made all data underlying the findings in their manuscript fully available?

Reviewer #1: Yes

Reviewer #2: Yes

Reviewer #4: Yes

5. Is the manuscript presented in an intelligible fashion and written in standard English?

Reviewer #1: Yes

Reviewer #2: Yes

Reviewer #4: Yes

6. Review Comments to the Author

Reviewer #1: (No Response)

Reviewer #2: Dear author

All comments have been addressed. The manuscript in this present form is acceptable for publication in the PLOS ONE journal.

Reviewer #4: The revised manuscript was good written and all reviewers' coments were addressed in the manuscript.

7. PLOS authors have the option to publish the peer review history of their article (what does this mean?). If published, this will include your full peer review and any attached files.

Reviewer #1: **Yes: **Holakoo Mohsenifar

Reviewer #2: **Yes: **Mehrnaz Kajbafvala

Reviewer #4: No
